# Truncated Expression of a Carboxypeptidase A from Bovine Improves Its Enzymatic Properties and Detoxification Efficiency of Ochratoxin A

**DOI:** 10.3390/toxins12110680

**Published:** 2020-10-29

**Authors:** Lu Xiong, Mengxue Peng, Meng Zhao, Zhihong Liang

**Affiliations:** 1College of Food Science and Nutritional Engineering, China Agricultural University, Beijing 100083, China; lxxionglu@163.com (L.X.); pengmx@cau.edu.cn (M.P.); zhaom@cau.edu.cn (M.Z.); 2The Supervision, Inspection and Testing Center of Genetically Modified Organisms, Ministry of Agriculture, Beijing 100083, China; 3Beijing Laboratory for Food Quality and Safety, College of Food Science and Nutritional Engineering, China Agricultural University, Beijing 100083, China

**Keywords:** truncated expression, Carboxypeptidase A, detoxification, Ochratoxin A

## Abstract

Ochratoxin A (OTA) is a toxic secondary metabolite produced mainly by *Penicillium* spp. and *Aspergillus* spp. and commonly found in foodstuffs and feedstuffs. Carboxypeptidase A (CPA) can hydrolyze OTA into the non-toxic product ochratoxin α, with great potential to realize industrialized production and detoxify OTA in contaminated foods and feeds. This study constructed a *P. pastoris* expression vector of mature CPA (M-CPA) without propeptide and signal peptide. The results showed that the degradation rate of OTA by M-CPA was up to 93.36%. Its optimum pH was 8, the optimum temperature was 40 °C, the value of K_m_ was 0.126 mmol/L, and the maximum reaction rate was 0.0219 mol/min. Compared with commercial CPA (S-CPA), the recombinant M-CPA had an improve stability, for which its optimum temperature increased by 10 °C and stability at a wide range pH, especially at pH 3–4 and pH 11. M-CPA could effectively degrade OTA in red wine. M-CPA has the potential for industrial applications, such as can be used as a detoxification additive for foods and feeds.

## 1. Introduction

Ochratoxin A (OTA) is a secondary metabolite of fungi such as *Aspergillus* spp. and *Penicillium* spp., with nephrotoxicity, teratogenicity, hepatotoxicity, and carcinogenicity, classified as a 2B carcinogen by the International Agency for Research on Cancer (IARC). OTA widely contaminates agricultural products like cereals [1], coffee [2], and wine [3]. The European Commission (EC/1881/2006) sets maximum levels of OTA in foodstuffs with the range of 0.5–10 μg/kg [4], and the National Food Safety Standard of China (GB2761-2017) also establishes the OTA limits for grains, beans, wine, and coffee in a range of 2–10 μg/kg [5]. The limits of OTA in feedstuffs are higher than those in foodstuffs. The Hygienical Standard for Feeds of China (GB13078-2017) stipulates an OTA limit of ≤100 μg/kg [6]. The European Commission has issued OTA limits for feed ingredients, and supplementary and compound feeds of pig and poultry at a range of 50–250 μg/kg (EC/576/2006) [7]. As one of the most deleterious mycotoxins, OTA is not easy to remove or destroy during food processing. To mitigate the health risks posed by the OTA contamination in foodstuffs and feedstuffs, several strategies are used to eliminate OTA in food and feed products, including physical methods and chemical methods. However, these methods usually damage nutrients and cause secondary contamination [8]. The biodegradation of OTA by enzymes is a very efficient, specific, and environmentally friendly way that would be a very promising strategy for the control of OTA in foods and feeds [9]. OTA degrading enzymes include carboxypeptidase, amidase, and protease [10,11,12]. Among them, carboxypeptidase A (CPA, EC 3.4.17.1), derived from bovine pancreas, was the first enzyme found to hydrolyze OTA in 1969 [10]. Subsequently, the enzymes from eukaryotic microorganisms were successively mined to degrade OTA. By screening the commercial hydrolases, Stander et al. found a crude lipase (Amano^TM^) derived from *A. niger* with OTA degrading ability [11]. Then, Abrunhosa et al. discovered the crude metalloenzyme of *A. niger* with OTA hydrolysis activity, and the degradation efficiency reached 99.8% at pH 7.5 [13,14]. Dobritzsch et al. isolated and purified one amidase from the crude lipase of *A. niger* to degrade OTA, and determined its crystal structure at 2.2 Å resolution [12]. In recent years, carboxypeptidases of degrading OTA were found in bacteria such as *Bacillus amyloliquefaciens* [15], *Acinetobacter* [16], and *B. subtilis* [17]. Compared with the other OTA degrading enzymes, CPA is an enzyme with rich crystal structures and a clear catalytic mechanism. It can hydrolyze the amide bond of OTA and produce the nontoxic ochratoxin α (OTα) and L-α-phenylalanine (Phe). So, CPA has the application prospect of removing OTA in food and feed products.

CPA was discovered in 1929 [18] and the first crystal of CPA was obtained from *Bos taurus* pancreas in 1937 [19]. It is a 35 kDa zinc-dependent metal carboxypeptidase, routinely obtained at an industrial scale from porcine or bovine pancreas [20]. It can cut off the C-terminal amino acid (especially aromatic amino acid) residues, but not proline, hydroxyproline, arginine, and lysine residues. CPA is highly specific for excising aromatic amino acids residues from peptides and proteins with a preference for phenylalanine [21]. In addition, the higher hydrolysis rates of peptide were observed in histidine and threonine rather than other amino acids such as serine and aspartic acid. In addition to its natural substrates, CPA also hydrolyzes esters with similar configurations, often with faster reaction rates [22]. This enzyme is naturally synthesized in the form of zymogen with a 12 kDa N-terminal propeptide that covers the catalytic pocket of the enzyme and keeps it inactive until removal of the propeptide by trypsin. Commercial CPA is mainly derived from pancreatic tissues such as bovines and pigs, with high extraction cost. In addition, natural CPA has poor thermal stability, which limits its use in food, medicine, and chemical industry. Therefore, there is a need to obtain a large amount of carboxypeptidase A with high stability and catalytic efficiency in vitro. The CPA zymogen was successfully expressed in *Pichia pastoris* by Shi et al., with a protein product of 150 mg/L. However, this zymogen needs to be hydrolysis with trypsin to generate mature CPA, then it was used to degrade OTA, with a degradation rate of 72.3% [23]. Can the mature peptide of CPA be expressed directly to omit the step of trypsin cleavage? The propeptide usually acts as an intramolecular chaperone, helping the target protein to fold correctly, and has an important influence on the structure and function of the protein. In the absence of the propeptide, the target protein usually cannot be expressed ideally. For example, when deletion of the propeptide of the lipase from *Rhizopus oryzae* occurs, and the mature peptide in *Saccharomyces cerevisiae* is expressed, the expression product has no lipase activity, while when expression with the propeptide occurs, the expression product has lipase activity [24]. However, the lipase derived from *R. chinensis* is different. The propeptide has no significant effect on the main enzymatic properties of *R. chinensis* lipase [25]. The expression of different truncated sequences (removal of the leader peptide or signal peptide) will have a great impact on the expression, activity, and stability of the protein. So, the expression of the mature peptide of CPA may retain the catalytic activity and stability of the enzyme.

This paper was aiming to obtain recombinant mature Carboxypeptidase A (M-CPA), which has the ability to degrade OTA and improves its thermal and acid–base stability. The propeptide of proCPA was truncated to obtain the mature CPA gene, and the gene was cloned and expressed in *P. pastoris* GS115 strain, then a 6×His tag was added to the N-terminus of the M-CPA to facilitate purification. Its enzymatic characteristics were tested, including optimal temperature, thermal tolerance, optimal pH, Michaelis constant K_m_ and V_max_, and its ability to detoxify OTA in vitro and red wine was evaluated. The enzymatic activity of M-CPA was compared to commercial CPA (S-CPA) from Sigma (St. Louis, MO, USA).

## 2. Results

### 2.1. Construction and Expression of M-CPA in P. pastoris

To achieve the expression of mature CPA with the ability to cleave OTA directly, M-CPA was constructed by deleting the 110 residues (12 kDa) including the signal peptide and propeptide. The amino acid sequences of the CPA and its truncated protein constructed in this study were summarized in Figure 1a. The CPA zymogen contains 417 amino acid residues (47 kDa), including a signal peptide composed of 16 amino acid residues, a leader peptide composed of 94 amino acid residues, and a mature peptide composed of 307 amino acid residues (35 kDa). The protein secretion by the recombinant *P. pastoris* GS115/pPIC9K/M-CPA reached 303.08 mg/L. The recombinant M-CPA containing six histidine residues in its N-terminal region was then purified using metal affinity chromatography on a Ni-NTA resin column. Each of these histidine-tagged carboxypeptidases bound to the Ni-NTA resin was eluted with imidazole. The remaining imidazole from the elution was removed by dialysis against 50 mM Tris–HCl (pH 7.5). After protein purification, bands at approximately 35 kDa were obtained by SDS-PAGE (Figure 1b) and Western blotting (Figure 1c), respectively. The purified proteins were used for further biochemical characterization.

### 2.2. M-CPA Improved Thermal Stability and Acid-Base Stability

Optimum temperature and thermal tolerance were used to characterize the thermal stability of M-CPA and S-CPA. The optimum temperature of S-CPA (purchased from Sigma) and M-CPA (truncated expression) were determined during assay with 25 mM Tris-HCl buffer and 500 mM sodium chloride hippuryl-L-phenylalanine (HLP) (pH 7.5) as the substrate (Figure 2a). The optimum temperatures of the S-CPA and M-CPA were found to be 30 °C and 40 °C, respectively (Figure 2a). This indicated that the thermal stability of M-CPA was higher than that of S-CPA. The thermal tolerance of the M-CPA and S-CPA was carried out by incubating at different temperatures (30–80 °C) for 10 min (Figure 2b). M-CPA and S-CPA retained 60.25% and 71.4% residual activity at a temperature of 40 °C for 10 min, respectively. Above 50 °C, the thermal tolerance of M-CPA was slightly better than that of S-CPA. However, their activity decreased more than 60% at temperatures above 50 °C and the residual activity only maintained 10% when the temperature was above 70 °C. These results suggested that the thermal stability of M-CPA and S-CPA were both poor.

Various buffers at different pH values were used to evaluate the optimum pH of the S-CPA and M-CPA. The optimum pH values for the enzymatic activity of both were found to be 8.0, and M-CPA showed high enzymatic activity within the pH range of 5–8. Interestingly, the M-CPA exhibited higher relative activity than S-CPA when assayed in pH 3–4, and there were no activities of S-CPA detected at pH below 4.0 (Figure 2c). The activity of both enzymes decreased significantly at pH > 8. S-CPA had only 28.54% relative activity at pH = 11, while recombinant M-CPA had 60.8%, indicating that M-CPA was more resistant to alkali than S-CPA.

Kinetic parameters of both enzymes were obtained by measuring the rates of hydrolysis of HLP at the concentration of 0.2, 0.4, 0.6, 0.8, and 1 mmol/L. K_m_ and V_max_ values were estimated by the Michaelis–Menten model. A Lineweaver–Burk plot of 1/V_0_ against 1/[S] was made and the regression equations of M-CPA and S-CPA were y = 5.77x + 45.64 and y = 3.23x + 31.70, respectively (Figure 2d). The Michaelis constant K_m_ values of M-CPA and S-CPA hydrolyzing HLP were 0.126 mmol/L and 0.102 mmol/L, respectively; the maximum rates (Vmax) of HLP hydrolysis were 0.0219 mol/min and 0.0315 mol/min, respectively. These results indicated that the substrate affinity of M-CPA for HLP was slightly lower than that of S-CPA. The deletion of propeptide of CPA at the N-terminal region has a marginal effect on the specific activity.

### 2.3. Detoxification of OTA by M-CPA

To determine the detoxification rate of M-CPA against OTA, the concentration of OTA was detected by HPLC. The HPLC chromatograms of degradation products of OTA were shown in Figure 3a. The retention times (RT) of OTA and its degradation product was 5.50 min and 2.25 min, respectively. After the treatment of OTA with S-CPA and M-CPA, the peak area of OTA decreased significantly compared with the control group, and the new product appeared at 2.25 min. According to the standard curve of OTA, the detoxification rates of S-CPA and M-CPA were 96.04% and 93.36%, respectively. It has been proved that S-CPA can degrade OTA to OTα [26]. Therefore, the substance appearing at 2.25 min after S-CPA+OTA treatment in Figure 3a should be OTα. Then, the presence of OTα was confirmed by LC-MS/MS analysis. The degradation products of OTA by M-CPA indicated the presence of a peak with a retention time of 4.36 min that had the same mass transition characteristics (m/z 255.0–211.0) of OTα. These data unequivocally identified OTα as the metabolite that resulted from OTA biodegradation by M-CPA.

### 2.4. M-CPA Effectively Degraded OTA in Red Wine

Because of the high incidence and levels of OTA in wines, it is of great significance to remove OTA from wines [27]. Some surveys showed that red wine generally contains a higher OTA amount than white and rosé wine [28]. So, we evaluated the ability of M-CPA to detoxify OTA in red wine. The biodegradation rate of OTA by M-CPA was significantly higher than that of S-CPA in red wine. Figure 4 showed the concentration of OTA and OTα detected by thin layer chromatography (TLC) and HPLC. The levels of OTA were reduction about 15.7% and 64% after incubation with S-CPA and M-CPA 24 h, respectively. S-CPA losing the ability to degrade OTA in red wine may be due to the presence of ethanol (13.6%) and the low pH (3.54), which may have inhibited the enzyme responsible for the hydrolysis of OTA.

## 3. Discussion

A wide range of food and feed products are contaminated by OTA[1,2,27]. The products that exceed the maximum limits of OTA can only be discarded. The economic loss caused by this is immeasurable. However, a considerable portion of them can be degraded into non-toxic products by biodegradation methods and then put into the market. Enzymatic degradation is considered to be the safest and most effective detoxification method. As an enzyme with clear crystal structures and catalytic mechanisms and capable of degrading OTA [29], CPA is of great significance for the detoxification of OTA in food and feed. However, CPA used in industry is mainly extracted from animals (such as pancreatic tissues of cattle and pigs), which has a relatively high cost, and its thermal stability and acid–base stability are poor. CPA from *Rattus norvegicus* lost 50% of its enzyme activity at 30 °C for 10 min [30], and bovine CPA was completely inactivated at 55 °C for 30 min [31]. So, natural CPA is not suitable for industrial applications. This paper expressed the mature peptide of CPA in *P. pastoris* GS115 strain, and the recombinant M-CPA had improved thermal stability and acid–base stability. The M-CPA could effectively degrade OTA in red wine.

The zymogen of CPA from bovine pancreatic was successfully expressed in *P. pastoris*, and the expressed product has OTA degradation activity after being hydrolyzed by trypsin [23]. However, the use of trypsin to cleave the CPA zymogen causes operational complexity and high cost. In order to express CPA that can directly degrade OTA and improve its stability, the truncated CPA was expressed as mature peptide in *P. pastoris*. By analyzing the crystal structure of CPA and carrying out molecular dynamics simulation, it was found that the N-terminal sequence of CPA was more flexible, which was not conducive to the stability of the protein. So, we truncated the 110 amino acid residues at the N-terminal, including 11 amino acid residues of the signal peptide and 94 amino acid residues of the propeptide (Figure 1a). Compared with S-CPA, the optimum temperature of M-CPA was increased by 10 °C (Figure 2a). Although their activity decreased more than 60% at temperatures above 50 °C for 10 min, the thermal tolerance of M-CPA was slightly better than that of S-CPA (Figure 2b). Combining the results of the optimum temperature and thermal tolerance experiments, the thermal stability of M-CPA was better than that of S-CPA. The optimum pH of M-CPA and S-CPA were both at 8.0, while M-CPA has better acid–base stability, especially at pH 3–4 and pH 11 (Figure 2c). These results indicated that M-CPA was more suitable for industrial production. The OTA detoxification rates of S-CPA and M-CPA were 96.04% and 93.36%, respectively (Figure 2a). In addition, both of them can degrade OTA into OTα (Figure 3b). M-CPA has a high degradation rate of OTA, compared with that of CPA zymogen, which has a degradation rate of 72.3% and needs to undergo hydrolysis by trypsin to generate mature CPA before reaction [23]. Compared with carboxypeptidases derived from bacteria, M-CPA also has a higher degradation rate. The OTA degradation rates of carboxypeptidases from *B. amyloliquefaciens* ASAG1 [15], *Acinetobacter* sp. neg1 [16], and *B. subtilis* CW14[17] were just 72%, 33%, and 71.3%, respectively. 

To test the application potential of M-CPA, we selected a red wine system and a simulated gastrointestinal digestion model to study the degradation of OTA by M-CPA. OTA was reduced about 15.7% and 64%, respectively, after incubation with S-CPA and M-CPA in red wine for 24 h (Figure 4). S-CPA has poor OTA degradation activity, which may be due to the lower pH (3.54) in red wine. This also reflected the acid–base stability of M-CPA. In the simulated gastrointestinal digestion process, the degradation rates of OTA by S-CPA and M-CPA were 20.80% and 71.29%, respectively (Appendix A). This indicated that M-CPA can tolerate acid, alkali, pepsin, and pancreatin in the gastrointestinal digestive environment. Due to its excellent stability, M-CPA has the potential for industrial applications, such as be used as a detoxification additive for foods and feeds or as a biological leaching material for wine to remove OTA.

## 4. Conclusions

To express CPA that can directly degrade OTA and improve its thermal stability and acid–base stability, this paper expressed the mature CPA in *P. pastoris*. The degradation rate of OTA by M-CPA up to 93.36%. Compared with S-CPA, the recombinant M-CPA had an improved stability, for which its optimum temperature increased by 10 °C and stability at a wide range pH, especially at pH 3–4 and pH 11. The M-CPA could effectively degrade OTA in red wine. M-CPA may be used as a detoxification additive of foods and feeds.

## 5. Materials and Methods

### 5.1. Chemicals and Strains

The strains *P. pastoris* GS115 and *Escherichia coli* HB101 were maintained in our lab. Commercial Carboxypeptidase A (S-CPA, derived from bovine pancreas, C9268-2.5KU), OTA (purity degree 98%), and hippuryl-L-phenylalanine were purchased from Sigma (St. Louis, MO, USA). The dry red wine was obtained from the Key Laboratory of Viticulture and Vitiology, Ministry of Agriculture of China. The grape variety used to make this wine was Cabernet Sauvignon, producing in Helanshan, Ningxia (38°34′ N, 106°20′ E). The wine was brewed in 2017, with an alcohol content of 13.4% (*v*/*v*) and a pH of 3.54.

### 5.2. Construction of the Expression Plasmid ppic9k/M-cpa

The native CPA gene from bovine (GenBank Accession No. NM_174750; CAA83955.1) without signal peptidase and truncated propeptide sequences was optimized based on the *P. pastoris* codon usage to reach the highest protein expression, then it was cloned into pPIC9K vector at EcoRⅠand NotⅠsites. The recombinant vector named pPIC9K/M-CPA was then transformed into an *E. coli* HB101 strain. All of this was synthesized by AuGCT (Beijing, China).

### 5.3. Transformation and Colony Screening of P. pastoris

The recombinant plasmid DNA was purified from *E. coli* HB101 using the EasyPure HiPure Plasmid MaxiPrep Kit from TransGen Biotech (Beijing, China). The recombinant plasmids were linearized by Sac I, then transformed into *P. pastoris* GS115 via electroporation (1.5 kV, 4 ms). Transformants were screened on minimal dextrose (MD) media (1.34% yeast nitrogen base without amino acids, 1% glycerol, 4 × 10^−5^ % biotin) plates. The pPIC9K without the insert was used as a negative control. The high expression transformants were screened with Peptone dextrose (YPD) medium plates (10 g/L yeast extract, 20 g/L tryptone, 20 g/L dextrose). Single His+ transformants were, respectively, inoculated into each microtiter well containing 200 μL YPD at 30 °C for 48 h, then 10 μL cultures from each well were correspondingly transformed into clones selected based on their ability to grow on YPD transferred into a new microtiter plate well containing 190 μL YPD. After incubation overnight at 30°C, the previous steps were repeated to create a third microtiter plate. The cells in each well were then resuspended and 10 μL from each well, in order, and were spotted on YPD plates containing geneticin G418 at a final concentration of 0, 1.0, 2.0, 3.0, and 4.0 mg mL. The YPD-geneticin plates were incubated at 30°C for 2–5 days for differentiating the high expression recombinants. The selected yeast transformants were confirmed by PCR and DNA sequencing.

### 5.4. Fermentation Conditions

The yeast fermentation in a 100 mL shaking flask was carried out according to the *P. pastoris* expression manual. A recombinant yeast colony was inoculated in 20 mL of buffered minimal glycerol/methanol-complex (BMGY/BMMY) medium (10 g/L yeast extract, 20 g/L tryptone, 10 g/L ammonium sulfate, 3.4 g/L yeast nitrogen base without amino acids and ammonium sulfate, 100 mM potassium phosphate buffer (pH6.0), 400 μg/L biotin,10 g/L glycerol or 5 g/L methanol). After culture at 28 °C for 48 h, the cells were collected by centrifugation and re-suspended in 20 mL of BMMY medium. The initial optical density value of culture at 600 nm was adjusted to 1.0. The incubation temperature was set at 28 °C, the rotation speed was 250 rpm. The methanol with a working concentration of 5 g/L was feed at 0, 24, 48, 72, 96, 120, and 144 h. During the induction course, supernatants samples were collected at 0, 24, 48, 72, 96, 120, 144, and 168 h. Protein expression in samples was analyzed using SDS-PAGE and Western blotting.

### 5.5. Purification of the Recombinant M-CPA

A 6×His tag was introduced to the N-terminus of the target protein. Thus, the recombinant M-CPA was expressed as a His-tagged soluble protein. The cell-free culture supernatant was collected by centrifugation at 6000× *g* for 10 min at 4 °C and was desalted using Amicon Ultra-15 Centrifugal Filter Devices (Millipore; 10 kDa MWCO). The protein was purified by Ni^2+^-affinity chromatography as described in the Ni-NTA resin manual from TransGen Biotechnology, Previously equilibrated with buffer A (pH 8.0 50 mM NaH2PO4, 300 mM NaCl and 10 mM imidazole). The mixture of 10 mL of the supernatant fraction containing the soluble recombinant CPA and resin was incubated for 15 min at 4 °C and applied to an empty column. The column was washed several times with buffer B (pH 8.0, 50 mM NaH_2_PO_4_, 300 mM NaCl and 20 mM imidazole). Finally, the recombinant proteins were eluted with buffer C (pH 8.0, 50 mM NaH_2_PO_4_, 300 mM NaCl and 250 mM imidazole). Elutes with protein were concentrated using Amicon Ultra-15 Centrifugal Filter Devices (Millipore; 10 kDa MWCO), and then lyophilized by vacuum freezing dryer.

### 5.6. SDS-PAGE and Western Blot Analysis

SDS-PAGE and Western blot analysis was performed using 12% gel by previously described methods [32]. Proteins were determined by Bradford protein assay kit and bovine serum was used as standard. The first antibody was mouse anti His antibody, and the second antibody was horseradish peroxidase (HRP)-labeled Goat Anti-Mouse IgG. An Immobilon Western Chemiluminescent HRP Substrate Kit was used to stain the target protein band.

### 5.7. Enzymatic Activity Measurements

The enzyme activity determination refers to the method of Tardioli et al. [33]. The initial reaction rate was measured using hippuryl-L-phenylalanine (Sigma) as a substrate. The assay mixture contained the recombinant M-CPA dissolved in the 25 mM Tris-HCl buffer and 500 mM sodium chloride Hippuryl-L-phenylalanine (pH 7.5). Mixing was done by inversion and the absorbance at 254 nm for approximately 5 min was recorded in a spectrophotometer. The fastest linear rate (ΔA254 nm/min) over a 30 s interval for the test and the blank reactions was obtained. The initial enzymatic reaction rate was estimated from the linear region of the absorbance versus time curve. One unit of the enzymes will hydrolyze 1.0 μmole of hippuryl-L-phenylalanine per min at pH 7.5 at 25 °C.

The optimal reaction pH was assessed using several buffers with varying pH values (50 mM acetic acid-sodium acetate buffer, pH 3.0–5.0; 50 mM NaHPO_4_-NaH_2_PO_4_ buffer, pH 6.0–7.5; 50 mM Tris-HCl buffer, pH 8.0–9.0; and 50 mM Gly-NaOH buffer, pH 9.0–12.0) at 25 °C.

The optimal temperature of M-CPA and S-CPA were determined using the optimal pH at temperatures ranging from 20 °C to 80 °C. The reaction solution was heated in a water bath at different temperature sfor 10min, and then the absorption value of this solution was tested by adding M-CPA or S-CPA. The calculation formula of relative enzyme activity was as follows: Relative activity = determined activity/activity at optimum temperature × 100%

The thermal tolerance assays were assessed by incubating the enzyme (M-CPA or S-CPA) at different temperatures (30 °C to 80 °C) for 10 min, and the non-heated enzyme (activity of enzyme in 25 °C with substrate) was used as the control (100%). The calculation formula of relative enzyme activity was as follows: Relative activity = determined activity/activity of non-heated enzyme × 100%.

K_m_ and V_max_ values were estimated by Michaelis–Menten model, according to the experimental kinetics data for substrate concentration at 0.2, 0.4, 0.6, 0.8, and 1 mmol/L. K_m_ and V_max_ can be obtained basing on the Lineweaver–Burk plot. It gives a straight line, with the intercept on the y-axis equal to 1/V_max_, and the intercept on the x-axis equal to the absolute value of 1/K_m_. The calculation formulas of K_m_ and V_max_ are as follows, where [S] refers to the concentration of the substrate.
$$\frac{1}{V}=\frac{K_{m}+\left[S\right]}{V_{\max}\left[S\right]}=\frac{K_{m}}{V_{\max}}\frac{1}{\left[S\right]}+\frac{1}{V_{\max}}$$

### 5.8. Ochratoxin A Degradation Assay

Degradation assays were done by incubating of recombinant M-CPA (5 U/mL) which dissolved in the 1 M NaCl (pH 7.5) with ochratoxin A solution at 37 °C for 24 h. The final concentration of ochratoxin A was 2 μg/mL. Assays with 5 U/mL of carboxypeptidase A (EC 3.4.17.1) and blanks without any enzyme were used as controls. The aqueous phase was acidified to pH = 2 and OTA was re-extracted three times with an equal volume of chloroform after centrifugation at 6000× *g* for 10 min, the organic phase was transferred to a clean tube, evaporated under nitrogen, and dissolved in 300 μL methanol. The remaining OTA in the supernatants were applied to thin layer chromatography (TLC) analysis and analyzed by high-performance liquid chromatography (HPLC).

#### 5.8.1. Thin Layer Chromatography Analysis

The method of TLC analysis was performed according to Shi et al. [23]. A total of 10 μL supernatants of OTA were spotted on silica gel plates. Toluene/ethyl acetate/formic acid (6:3:1 *v*/*v*/*v*) was used as the developing solvent. After 10 min of separating, OTA was chromogenic by the UV light of gel imaging analyzer.

#### 5.8.2. HPLC Analysis

HPLC analysis of OTA and OTα was carried out according to the method described by Hu [17]. Some of the HPLC parameters were different from the method as below: The concentration of OTA and OTα was evaluated with the Water Alliance 2695-2475 UPLC system using a C18 reversed-phase (150 × 4.6 mm and 3.5 mm). A five-point calibration curve (10; 250; 500; 750 and 1000 ng/mL) was prepared with standards of OTA (Sigma). The rate of OTA degradation was calculated using the formula: OTA degradation rate = (1 − OTA peak area in treatment/OTA peak area in control) × 100%.

### 5.9. LC/MS-MS Methods

OTA and OTα were analyzed by Q Exactive orbitrap mass spectrometer (Thermo, CA, USA). A Cortecs C18 100 × 2.1 mm 1.9 μm column was used for analysis. Mobile phase A was 2 mM ammonium acetate in 100% water. Mobile phase B was isopropanol and acetonitrile (90:10, v:v). A 15-min gradient with a flow rate of 220 μL/min from 5% to 98% mobile B was used.

The detailed mass spectrometer parameters are as follows: ion source, high resolution electrospray ionization (H-ESI); ion mode, anion; spray voltage, 3000 v; flow rate of sheath gas, 35 Arb; flow rate of auxiliary gas, 10 Arb; temperature of ion transfer tube, 320 °C; heater temperature, 300 °C; full scan resolution, 70,000; fragment resolution, 17,500.

### 5.10. Detoxification of OTA in Red Wine

To evaluate if M-CPA was able to biodegrade OTA in red wine, OTA was supplemented into red wine. OTA standard was added into 300 μL red wine samples with the final concentration of 2 μg/mL. There are three groups, containing an experimental group (5U of M-CPA), a control group (5U of S-CPA) and a blank group. After the tubes were shaken and evenly mixed, they were incubated in a 200 rpm/min shaker at 37 °C for 24 h. The detoxification effect was detected according to the method in Section 5.8.1, and the detoxification rate of recombinant CPA in wine samples was determined according to the method in Section 5.8.2.

### 5.11. Statistical Analysis

The activity of M-CPA and S-CPA were expressed as mean ± standard deviation of three independent experiments. The SPSS 20.0 statistical package was used to perform one-way analysis of variance (ANOVA), and Duncan’s multiple range test was used to determine the significantly different (*p* < 0.05) between multiple samples. The Origin 7.5 software was used for drawing.

## Figures and Tables

**Figure 1 toxins-12-00680-f001:**
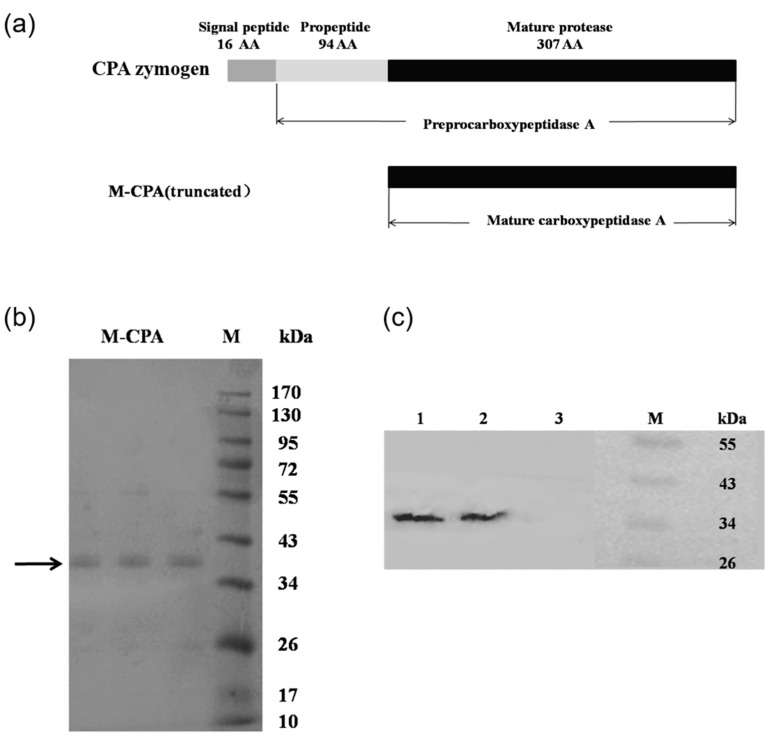
Expression of mature Carboxypeptidase A (M-CPA) in *P. pastoris*. (**a**) Schematic representation of the primary structures of CPA; (**b**) SDS-PAGE electrophoresis of recombinant M-CPA; (**c**) Western blot analysis of recombinant M-CPA.

**Figure 2 toxins-12-00680-f002:**
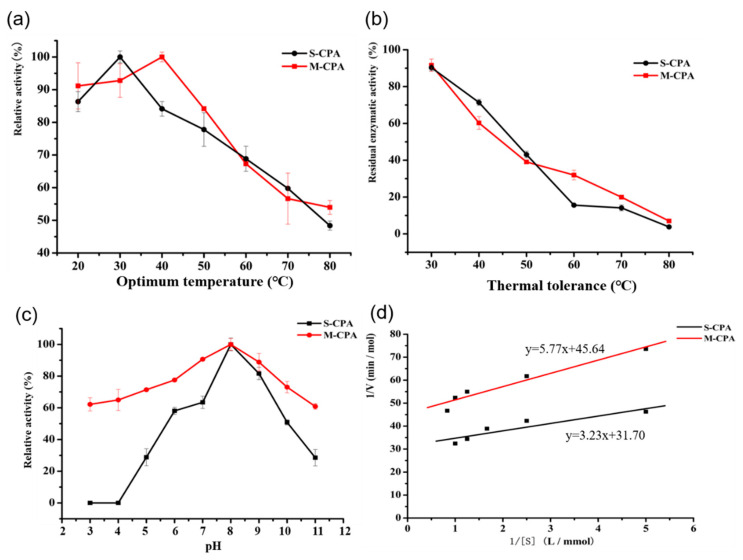
Enzymatic properties of recombinant M-CPA and commercial CPA (S-CPA). (**a**) The optimum temperature of M-CPA and S-CPA; (**b**) the thermal tolerance of M-CPA and S-CPA; (**c**) The optimum pH of M-CPA and S-CPA; (**d**) enzymatic dynamic response curve of M-CPA and S-CPA.

**Figure 3 toxins-12-00680-f003:**
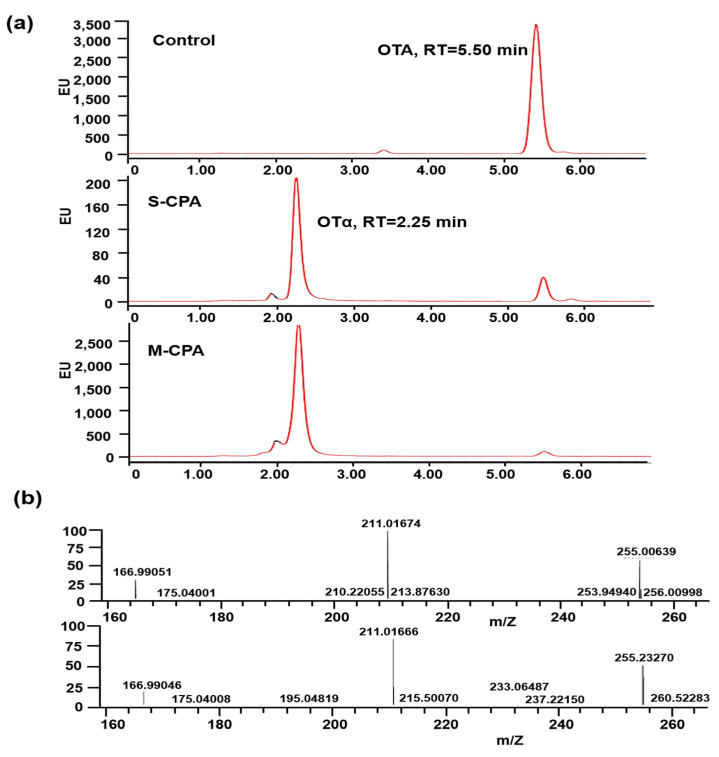
Determination of Ochratoxin A (OTA) and OTα by HPLC and LC-MS/MS, respectively. (**a**) HPLC chromatogram of degradation products of OTA after incubation with buffered minimal methanol-complex (BMMY) medium (control), S-CPA (ochratoxin α control), and M-CPA, respectively; (**b**) LC-MS/MS analysis of the degradation products of OTA incubation with S-CPA (above) and M-CPA (below), respectively.

**Figure 4 toxins-12-00680-f004:**
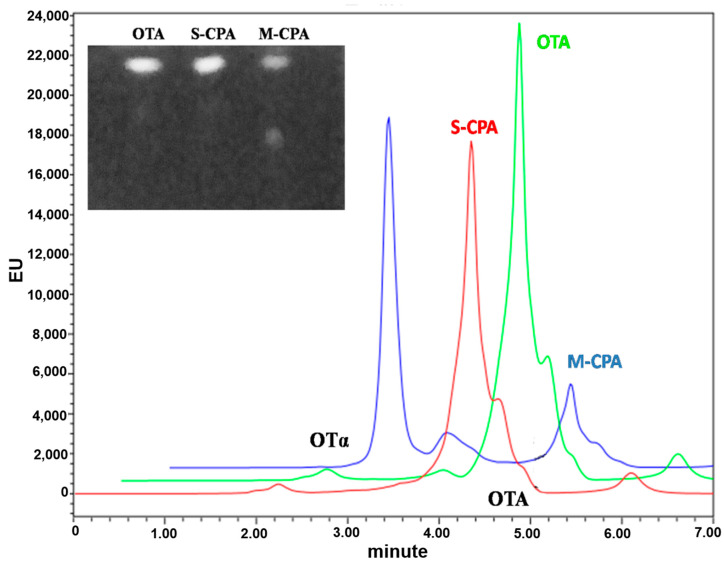
Thin layer chromatography (TLC) and HPLC analysis of OTA biodegradation in red wine by recombinant M-CPA and S-CPA.

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
