# Peer review of "Truncated Expression of a Carboxypeptidase A from Bovine Improves Its Enzymatic Properties and Detoxification Efficiency of Ochratoxin A"

_toxins, 2020, doi:10.3390/toxins12110680_

Round 1

Reviewer 1 Report

ROW 6. and 7.: Taxonomy should be writen in italyc.

ROW 10: What does M mean in M-CPA, cause it is only writen, that it is truncated? I suppose, it is mature?

ROW 10: What does S mean in S-CPA, cause it is only writen, that it is truncated?

The mention of the already identified and investigated CPA-s are missing form the Introduction and/or from the Discusion. There are a few CPA isolated from bacteriea's and fungi also, which are having quite interesting abilities and biodegradation potentials. I miss the comperison with those data’s also.

All the genetic information is missing from the investigated CPA. The sequences are missing, or it should be writen, that it is a protected data.

The size of the investigated CPA should be writen in total also. There is the propeptide, which is 35 kDa and there is the zymogene, which is 12 kDa, altogether it is 47 kDa.

The comperison/ or hint to the known CPA's in literature is missing.

The molecular sequence and the slicing points for separating the zymogene from the propeptide on the bp order shuold be also writen.

Are these sequences and Amino acid orders deposited in a genebank somewhere?

ROW 179-180: „we expresed the CPA with truncated propeptide” This sentence is not correct. This could be good: "the truncated CPA was expresed as propeptide".

Reference 22. (Hu et al., 2018) is only mentioned in the HPLC method, while they were identifing and isolating a new CPA peptide from Bacillus subtilis. This negligation of this important informations is strange.

Author Response

Manuscript ID: toxins-942214

The line numbers mentioned below refer to those in the revised manuscript. Point by point responses to editor/reviewers:

Reviewer #1:

We are very grateful to the Reviewer #1 for giving us very professional and specific suggestions to revise our manuscript.

Q1. ROW 6. and 7.: Taxonomy should be writen in italyc.

Thanks for the suggestion. It has been corrected in the text.

Q2. ROW 10: What does M mean in M-CPA, cause it is only writen, that it is truncated? I suppose, it is mature? What does S mean in S-CPA, cause it is only writen, that it is truncated?

Thanks for the suggestion. M-CPA means the recombinant mature Carboxypeptidase A.

S-CPA means the commercial CPA purchased from Sigma.

At line 9-10, we have revised the description to avoid ambiguity: This study constructed a P. pastoris expression vector of mature CPA (M-CPA) which without propeptide and signal peptide.

Q3. The mention of the already identified and investigated CPA-s are missing form the Introduction and/or from the Discussion. There are a few CPA isolated from bacteriea's and fungi also, which are having quite interesting abilities and biodegradation potentials. I miss the comperison with those data’s also.

Thanks for the suggestion. We totally agree with this comment and supplements the background of CPA derived from microorganisms at the Introduction and Discussion sections.

At line 40-49: OTA degrading enzymes include carboxypeptidase, amidase, and protease [10-12]. Among them, carboxypeptidase A (CPA, EC 3.4.17.1), derived from bovine pancreas, was the first enzyme found to hydrolyze OTA at 1969 [10]. Subsequently, the enzymes from eukaryotic microorganisms were successively mined to degrade OTA. By screening the commercial hydrolases, Stander et al found a crude lipase (Amano TM) derived from A. niger with OTA degrading ability [11]. Then, Abrunhosa et al discovered the crude metalloenzyme of A. niger with OTA hydrolysis activity, and the degradation efficiency reached 99.8% at pH 7.5 [13, 14]. Dobritzsch et al. isolated and purified one amidase from the crude lipase of A. niger to degrade OTA, and determined its crystal structure at 2.2 Å resolution [12]. In recent years, carboxypeptidases of degrading OTA was found in bacteria such as Bacillus amyloliquefaciens [15], Acinetobacter [16], and B. subtilis [17].

At line 196-203: M-CPA has a high degradation rate of OTA, comparing with that of CPA zymogen, which has a degradation rate of 72.3% and needs to be hydrolysis by trypsin to generate mature CPA before reaction [23]. Compared with carboxypeptidases derived from bacteria, M-CPA also has a higher degradation rate. The OTA degradation rates of carboxypeptidases from B. amyloliquefaciens ASAG1 [15], Acinetobacter sp. neg1 [16], and B. subtilis CW14[17] were just 72%, 33%, and 71.3% respectively.

Q4. All the genetic information is missing from the investigated CPA. The sequences are missing, or it should be writen, that it is a protected data.

Thanks for the suggestion. We have provided the sequence information at supplementary S2.

Q5. The size of the investigated CPA should be writen in total also. There is the propeptide, which is 35 kDa and there is the zymogene, which is 12 kDa, altogether it is 47 kDa.

Thanks for the suggestion. We supplement the description of CPA schematic representation.

At line 98-100: The CPA zymogen contains 417 amino acid residues (47 kDa), including a signal peptide composed of 16 amino acid residues, a leader peptide composed of 94 amino acid residues, and a mature peptide composed of 307 amino acid residues (35 kDa).

Q6. The comperison/ or hint to the known CPA's in literature is missing.

Thanks for the suggestion. We supplement the background of CPA derived from microorganisms at the Discussion sections.

At line 199-204: M-CPA has a high degradation rate of OTA, comparing with that of CPA zymogen, which has a degradation rate of 72.3% and needs to be hydrolysis by trypsin to generate mature CPA before reaction [23]. Compared with carboxypeptidases derived from bacteria, M-CPA also has a higher degradation rate. The OTA degradation rates of carboxypeptidases from B. amyloliquefaciens ASAG1 [15], Acinetobacter sp. neg1 [16], and B. subtilis CW14[17] were just 72%, 33%, and 71.3% respectively.

Q7. The molecular sequence and the slicing points for separating the zymogene from the propeptide on the bp order shuold be also writen. Are these sequences and Amino acid orders deposited in a genebank somewhere?

Thanks for the suggestion. We have provided the sequence information at supplementary S2. The native CPA gene from bovine can be found at GenBank (Accession No. NM_174750 ; GenBank: CAA83955.1)

Q8. W 179-180: „we expresed the CPA with truncated propeptide” This sentence is not correct. This could be good: "the truncated CPA was expresed as propeptide".

Thanks for your suggestion, we have revised it in the manuscript.

At line 185-186: In order to express CPA that can directly degrade OTA and improve its stability, the truncated CPA was expressed as mature peptide in P. pastoris.

Q9. Reference 22. (Hu et al., 2018) is only mentioned in the HPLC method, while they were identifing and isolating a new CPA peptide from Bacillus subtilis. This negligation of this important informations is strange.

Thanks for the suggestion. We supplement the background of CPA derived from microorganisms at the Discussion sections.

At line 196-203: M-CPA has a high degradation rate of OTA, comparing with that of CPA zymogen, which has a degradation rate of 72.3% and needs to be hydrolysis by trypsin to generate mature CPA before reaction [23]. Compared with carboxypeptidases derived from bacteria, M-CPA also has a higher degradation rate. The OTA degradation rates of carboxypeptidases from B. amyloliquefaciens ASAG1 [15], Acinetobacter sp. neg1 [16], and B. subtilis CW14[17] were just 72%, 33%, and 71.3% respectively.

Reviewer 2 Report

The manuscript entitled “Truncated expression of a carboxypeptidase A from bovine improves its enzymatic properties and detoxification efficiency of Ochratoxin A”, presented an investigation about biodegradation of ochratoxin A in a simulated gastrointestinal matrix and in a red wine sample, using a recombinant carboxypeptidase A. This work is interesting due to the use of a truncated sequence to obtain a recombinant mature carboxypeptidase, rather than the following detoxification experiments. However, different important points are missing, and some modifications are needed before it is accepted for possible publication. Different concerns about this manuscript are:

  1. The authors would explain the novelty of the results compared with previous works (Shi et al., 2015; Hu et al., 2018). This statement would be discussed in the introduction section more deeply. See, lines 53 to 60.
  2. The experimental design not compared adequately the results obtained, values of statistical analysis are missing, and no information about the number of replicate experiments was provided. Therefore, data are not robust enough to draw consistent conclusions.
  1. In view of the results, the authors would provide some hypothesis of the safety of its potential use in this food matrix (wine), and the limitations of this technique need to be addressed.
  2. The results of the study are not completely explained and discussed in the text. For instance, the discussion section should be rewritten comparing the results with previous biological detoxification studies with enzymes.
  3. In my opinion, some parts of the text are not properly organized, in the results section: the logic order of the subsections could be: 2.1, 2.3, 2.2, 2.5, and 2.4.

The following considerations may be taken into account by the authors:

- Lines 27-36: I believe that it would be necessary to insert the average contents and maximum limits of OTA in feed. The study of OTA reduction in the gastrointestinal model could be applied to detoxication of feed. This information could be very helpful and explanatory in the context of the introduction section.

- Lines 29-30: “(EC, 1881/2006)” and “(GB2761-2017)” should be cited as references and included in the references section.

- Line 70: “…with the ability to degrade OTA efficiently and stably” I don’t understand the real significance of “stably” in this sentence.

- Line 73: “…optimal pH, substrate specificity, and thermal stability…were tested” Please insert a clear reference about substrate specificity of the enzyme studied in the text and in Figure 3.

- Line 74: “and the ability to detoxify OTA in different contaminate foodstuffs was evaluated” This sentence is inaccurate because only one food matrix (red wine) was investigated in this study.

- Lines 92 to 103: I did not completely understand the goals of this results section (analytical identification or detoxification of standards solutions) or how this is done. I would be helpful to have a more detailed description of this part.

- Line 147: Please change “in vitro” by “in vitro”, and through the text.

- Line 159: Please insert a complete Table 1 caption. For instance: “Detoxification of OTA by M-CPA and S-CPA enzymes against control, during simulating gastrointestinal process”

- Line 177: The percentage calculus of the overall mean OTA reduction in the different matrices studied would be clearly explained in the materials and methods section.

- Line 206: “M-CPA may be used as a detoxification additive of food and feed”. This sentence is rather ambiguous and inaccurate. I would be more cautious in concluding positive results in different food types and feed, considering that main conclusions are drawn studying only one red wine sample and a simulated gastrointestinal model.

- Line 210: Because the importance of the OTA reduction data investigated in this work, it should be provided detailed information of the commercial OTA standard used.

- Line 276: Please change “1 mmol L-1” by “1 mmol L‒1

- Lines 285-288: “…were applied to high-performance TLC plates and analyzed by high-performance liquid chromatography (HPLC).” It is not clear how OTA solutions were analysed, because Figure 2 shows HPLC and LC-MS/MS chromatograms but not TLC plates.

- Lines 295-297: It is difficult to explore the behaviour of this M-CPA in different food matrices, only doing detoxication experiments in one matrix (red wine) at one fortification level (2 µg mL‒1). It is necessary to provide more information about the red wine used. For instance, a variety of grape, vintage date, finished wine or not, etc.

- References: The authors would check the citations according to the journal’s instructions.

Author Response

Manuscript ID: toxins-942214

The line numbers mentioned below refer to those in the revised manuscript. Point by point responses to editor/reviewers:

Reviewer #2:

We are very grateful to the Reviewer #1 for giving us very professional and specific suggestions to revise our manuscript.

Q1: The authors would explain the novelty of the results compared with previous works (Shi et al., 2015; Hu et al., 2018). This statement would be discussed in the introduction section more deeply. See, lines 53 to 60.

Thanks for the suggestion. We totally agree with this comment and supplements the background of CPA derived from microorganisms at the Introduction and Discussion sections.

At line 40-49: OTA degrading enzymes include carboxypeptidase, amidase, and protease [10-12]. Among them, carboxypeptidase A (CPA, EC 3.4.17.1), derived from bovine pancreas, was the first enzyme found to hydrolyze OTA at 1969 [10]. Subsequently, the enzymes from eukaryotic microorganisms were successively mined to degrade OTA. By screening the commercial hydrolases, Stander et al found a crude lipase (Amano TM) derived from A. niger with OTA degrading ability [11]. Then, Abrunhosa et al discovered the crude metalloenzyme of A. niger with OTA hydrolysis activity, and the degradation efficiency reached 99.8% at pH 7.5 [13, 14]. Dobritzsch et al. isolated and purified one amidase from the crude lipase of A. niger to degrade OTA, and determined its crystal structure at 2.2 Å resolution [12]. In recent years, carboxypeptidases of degrading OTA was found in bacteria such as Bacillus amyloliquefaciens [15], Acinetobacter [16], and B. subtilis [17].

At line 196-203: M-CPA has a high degradation rate of OTA, comparing with that of CPA zymogen, which has a degradation rate of 72.3% and needs to be hydrolysis by trypsin to generate mature CPA before reaction [23]. Compared with carboxypeptidases derived from bacteria, M-CPA also has a higher degradation rate. The OTA degradation rates of carboxypeptidases from B. amyloliquefaciens ASAG1 [15], Acinetobacter sp. neg1 [16], and B. subtilis CW14[17] were just 72%, 33%, and 71.3% respectively.

Q2. The experimental design not compared adequately the results obtained, values of statistical analysis are missing, and no information about the number of replicate experiments was provided. Therefore, data are not robust enough to draw consistent conclusions.

Thanks for the suggestion. We complement the statistical analysis.

At line 346-350:  5.12. Statistical analysis

The activity of M-CPA and S-CPA were expressed as mean ± standard deviation of three independent experiments. The SPSS 20.0 statistical package was used to perform one-way analysis of variance (ANOVA), and Duncan's multiple range test was used to determine the significantly different (p < 0.05) between multiple samples. The Origin 7.5 software was used for drawing.

Q3: In view of the results, the authors would provide some hypothesis of the safety of its potential use in this food matrix (wine), and the limitations of this technique need to be addressed.

Thanks for the suggestion. As the comments in the review published by Abrunhosa et al (2010), OTα (OTA degradation products) is essentially non-toxic. For example, OTα was ineffective as an immunosuppressor when tested in mice (Creppy et al,1983) and was considered 1,000-times less toxic than OTA in brain cells cultures (Bruinink et al,1998). Furthermore, OTα has an elimination half-live of 9.6 h in rats, which is well below that of OTA (103 h) (Li,1997). Therefore, processes that lead to the conversion of OTA into OTα contributes substantially to reduce OTA toxic effects and, hence, are considered to be routes for OTA detoxification.

Abrunhosa, L., Paterson, R. R. & Venancio, A. (2010) Biodegradation of ochratoxin a for food and feed decontamination, Toxins. 2, 1078-99.

Creppy, E.E.; Stormer, F.C.; Roschenthaler, R.; Dirheimer, G. Effects of two metabolites of ochratoxin A, (4R)-4-hydroxyochratoxin A and ochratoxin , on immune response in mice. Infect. Immun. 1983, 39, 1015–1018.

Bruinink, A.; Rasonyi, T.; Sidler, C. Differences in neurotoxic effects of ochratoxin A, ochracin and ochratoxin- in vitro. Nat. Toxins 1998, 6, 173–177.

Li, S.; Marquardt, R.R.; Frohlich, A.A.; Vitti, T.G.; Crow, G. Pharmacokinetics of ochratoxin A and its metabolites in rats. Toxicol. Appl. Pharmacol. 1997, 145, 82–90.

Q4: The results of the study are not completely explained and discussed in the text. For instance, the discussion section should be rewritten comparing the results with previous biological detoxification studies with enzymes.

Thanks for the suggestion. We totally agree with this comment and rewrite the discussion.

At line 190-203: Compared with S-CPA, the optimum temperature of M-CPA was increased by 10℃ (Fig.2 a). Although their activity decreased more than 60% at temperatures above 50 ℃ for 10 min, the thermal tolerance of M-CPA was slightly better than that of S-CPA (Fig.2 b). Combining the results of the optimum temperature and thermal tolerance experiments, the thermal stability of M-CPA was better than that of S-CPA. The optimum pH of M-CPA and S-CPA were both at 8.0, while M-CPA has better acid-base stability, especially at pH 3-4 and pH 11 (Fig.2 c). These results indicated that M-CPA was more suitable for industrial production. The OTA detoxification rates of S-CPA and M-CPA were 96.04% and 93.36% respectively (Fig.2a). And both of them can degrade OTA into OTα (Fig.3 b). M-CPA has a high degradation rate of OTA, comparing with that of CPA zymogen, which has a degradation rate of 72.3% and needs to be hydrolysis by trypsin to generate mature CPA before reaction [23]. Compared with carboxypeptidases derived from bacteria, M-CPA also has a higher degradation rate. The OTA degradation rates of carboxypeptidases from B. amyloliquefaciens ASAG1 [15], Acinetobacter sp. neg1 [16], and B. subtilis CW14[17] were just 72%, 33%, and 71.3% respectively.

Q5. In my opinion, some parts of the text are not properly organized, in the results section: the logic order of the subsections could be: 2.1, 2.3, 2.2, 2.5, and 2.4.

Thanks for your suggestion. We totally agree with this comment and revised it in the manuscript.

Q6. Lines 27-36: I believe that it would be necessary to insert the average contents and maximum limits of OTA in feed. The study of OTA reduction in the gastrointestinal model could be applied to detoxication of feed. This information could be very helpful and explanatory in the context of the introduction section.

Thanks for the suggestion. We have supplemented the OTA limits for feeds in the EU and China.

At line 27-33: The European Commission (EC/1881/2006) sets maximum levels of OTA in foodstuffs with the range of 0.5–10 μg/kg [4], and the National Food Safety Standard of China (GB2761-2017) also establishes the OTA limits for grains, beans, wine, and coffee in a range of 2-10 μg/kg [5]. The limits of OTA in feedstuffs is higher than those in foodstuffs. The Hygienical Standard for Feeds of China (GB13078-2017) stipulates a OTA limit of ≤100 μg/kg [6]. The European Commission has issued OTA limits for feed ingredients, and supplementary and compound feeds of pig and poultry at a range of 50-250 μg/kg (EC/576/2006) [7].

Q7. Lines 29-30: “(EC, 1881/2006)” and “(GB2761-2017)” should be cited as references and included in the references section.

Thanks for your suggestion. We revised it in the manuscript.

Q8. Line 70: “…with the ability to degrade OTA efficiently and stably” I don’t understand the real significance of “stably” in this sentence.

Thanks for your suggestion. We rewrite this sentence.

At line 82-83: This paper was aiming to obtain recombinant mature Carboxypeptidase A (M-CPA), which has the ability to degrade OTA and improves its thermal and acid-base stability.

Q9.- Line 73: “…optimal pH, substrate specificity, and thermal stability…were tested” Please insert a clear reference about substrate specificity of the enzyme studied in the text and in Figure 3.

Thanks for your suggestion. We have supplemented the reference of enzyme activity determination methods.

At line 284: The enzyme activity determination refers to the method of Tardioli et al.

Tardioli, P. W., Fernandez-Lafuente, R., Guisan, J. M. & Giordano, R. L. C. (2003) Design of new immobilized-stabilized carboxypeptidase A derivative for production of aromatic free hydrolysates of proteins, Biotechnology Progress. 19, 565-574.

Q10- Line 74: “and the ability to detoxify OTA in different contaminate foodstuffs was evaluated” This sentence is inaccurate because only one food matrix (red wine) was investigated in this study.

Thanks for your suggestion. We rewrite this sentence.

At line 86-88: Its enzymatic characteristics, including optimal temperature, optimal pH, substrate specificity, and thermal stability were tested, and the ability to detoxify OTA in vitro and red wine was evaluated.

Q11- Lines 92 to 103: I did not completely understand the goals of this results section (analytical identification or detoxification of standards solutions) or how this is done. I would be helpful to have a more detailed description of this part.

Thanks for your suggestion. We rewrite the methods of ochratoxin A degradation assay.

At line 309-328:

5.8. Ochratoxin A Degradation Assay

Degradation assays were done by incubating of recombinant M-CPA (5 U/ml) which dissolved in the 1 M NaCl (pH 7.5) with ochratoxin A solution at 37°C for 24 h. The final concentration of ochratoxin A was 2 μg/mL. Assays with 5 U/mL of carboxypeptidase A (EC 3.4.17.1) and blanks without any enzyme were used as controls. The aqueous phase was acidified to pH=2 and OTA was re-extracted three times with an equal volume of chloroform after centrifugation at 6000×g for 10 min, the organic phase was transferred to a clean tube, evaporated under nitrogen, dissolved in 300 μL methanol. The remaining OTA in the supernatants were applied to thin layer chromatography (TLC) analysis and analyzed by High-performance liquid chromatography (HPLC).

5.8.1 Thin layer chromatography analysis

The method of TLC analysis was performed according to Shi et al [23]. 10 μL supernatants of OTA were spotted on silica gel plates. Toluene/ethyl acetate/formic acid (6:3:1 v/v/v) was used as developing solvent. After 10 min of separating, OTA was chromogenic by the UV light of gel imaging analyzer.

5.8.2 HPLC analysis

HPLC analysis of OTA and OTα was carried out as the method described by Hu [17]. Some of the HPLC parameters were different from the method as below: The concentration of OTA and OTα was evaluated with the Water Alliance 2695-2475 UPLC system using a C18 reversed-phase (150×4.6 mm and 3.5 mm). A five-point calibration curve (10; 250; 500; 750 and 1000 ng/mL) was prepared with standards of OTA (Sigma). The rate of OTA degradation was calculated using the formula: OTA degradation rate = (1 − OTA peak area in treatment /OTA peak area in control) × 100 %

Q12:- Line 147: Please change “in vitro” by “in vitro”, and through the text.

Thanks for your suggestion. We revised it in the manuscript.

Q13. Line 159: Please insert a complete Table 1 caption. For instance: “Detoxification of OTA by M-CPA and S-CPA enzymes against control, during simulating gastrointestinal process”

Thanks for your suggestion. We revised it in the manuscript. These data have been adjusted to supplementary materials.

Q14.- Line 177: The percentage calculus of the overall mean OTA reduction in the different matrices studied would be clearly explained in the materials and methods section.

Thanks for your suggestion. We revised it in the manuscript.

At line 327-328: The rate of OTA degradation was calculated using the formula: OTA degradation rate = (1 − OTA peak area in treatment /OTA peak area in control) × 100 %

Q15.  Line 206: “M-CPA may be used as a detoxification additive of food and feed”. This sentence is rather ambiguous and inaccurate. I would be more cautious in concluding positive results in different food types and feed, considering that main conclusions are drawn studying only one red wine sample and a simulated gastrointestinal model.

Thanks for your suggestion. We revised it in the manuscript.

At line 212-214: Due to its excellent stability, M-CPA has the potential for industrial applications, such as be used as a detoxification additive for foods and feeds or as a biological leaching material for wine to remove OTA.

Q16.  Line 210: Because the importance of the OTA reduction data investigated in this work, it should be provided detailed information of the commercial OTA standard used.

Thanks for your suggestion, we have provided the information in the 5.1. Chemicals and Strains section.

At line 224-225: Commercial Carboxypeptidase A (S-CPA, derived from bovine pancreas, C9268-2.5KU)

Q17.- Line 276: Please change “1 mmol L-1” by “1 mmol L‒1

Thanks for your suggestion. We have revised it in the manuscript.

Q18- Lines 285-288: “…were applied to high-performance TLC plates and analyzed by high-performance liquid chromatography (HPLC).” It is not clear how OTA solutions were analysed, because Figure 2 shows HPLC and LC-MS/MS chromatograms but not TLC plates.

Thanks for your suggestion. We have supplemented the method of TLC.

At line 317-321:

5.8.1 Thin layer chromatography analysis

The method of TLC analysis was performed according to Shi et al [23]. 10 μL supernatants of OTA were spotted on silica gel plates. Toluene/ethyl acetate/formic acid (6:3:1 v/v/v) was used as developing solvent. After 10 min of separating, OTA was chromogenic by the UV light of gel imaging analyzer.

Q19- Lines 295-297: It is difficult to explore the behaviour of this M-CPA in different food matrices, only doing detoxication experiments in one matrix (red wine) at one fortification level (2 µg mL‒1). It is necessary to provide more information about the red wine used. For instance, a variety of grape, vintage date, finished wine or not, etc.

Thanks for your suggestion. We have supplemented the method of Detoxification of OTA in red wine.

At line 339-345: 5.10. Detoxification of OTA in red wine

To evaluate if M-CPA was able to biodegrade OTA in red wine, OTA was supplemented into red wine. OTA standard was added into 300 μL red wine samples with the final concentration of 2 μg/mL. There are three groups, containing experimental group (10mg S-CPA), control group (5U S-CPA) and blank group. After the tubes were shaken and evenly mixed, they were incubated in a 200 rpm /min shaker at 37 ℃ for 24 h. The detoxification effect was detected according to the method in section 5.8.1, and the detoxification rate of recombinant CPA in wine samples was determined according to the method in section 5.8.2.

At line 226-228: The red wine was obtained from the Key Laboratory of Viticulture and Vitiology, Ministry of Agriculture of China. The ethanol content and pH of the red wine was 13.4% (v/v) and 3.54 respectively.

Q20- References: The authors would check the citations according to the journal’s instructions.

Thanks for your suggestion. We revised it in the manuscript.

Reviewer 3 Report

General comments

The manuscript presented the expression of mature carboxypeptidase A (M-CPA) by Pichia pastoris GS115 transformed with a truncated CPA gene. The enzyme activities of M-CPA and detoxification of Ochratoxin A (OTA) by M-CPA were studied under various conditions using carboxypeptidase A purchased from Sigma (S-CPA) as a reference. M-CPA remained more than 60% activity in the pH range from 3 to 11 and effectively detoxified OTA in red wine and in a simulated gastrointestinal digestion system while S-CPA only worked in the narrow pH range and showed low detoxification activity . Although the data showed that the optimum temperature for M-CPA increased by 10℃ (from 30 to 40℃) compared with S-CPA, the enzyme activity of M-CPA reduced by 40% after incubating at 40℃ for 10 m. So the improvement of thermal stability by M-CPA is questionable. It seems data were not analyzed statistically.

Specific comments

The data of M-CPA production rate (efficiency) by P. pastoris should be added into the section 2.1 right after Fig. 1 as this information is important to the up-scale enzyme production.

Line 112: How was the optimum temperature obtained? Please write detail method in the method section.

Line 114-116: “The M-CPA and S-CPA retained 60.25% and 71.4% residual activity at temperature 40℃ for 10 min, respectively.” Why did this indicate that the thermal stability of M-CPA was higher than that of S-CPA?

Line 121-122: “the M-CPA showed optimum pH within the range of 5–8.” should be changed to the M-CPA showed high enzymatic activity within the pH range of 5–8.

Line 160: is it table1 ? The data in the table should be statistically analyzed.

Line 209: Please change “stored” to “maintained “

Line274: Please specify the temperature for “the non-heated enzyme”.

Line 309: Just after line309, please add method for detoxification in red wine.

Line 318: Just after line 318, please add statistic data analysis

Author Response

[Toxins] Manuscript ID: toxins-942214

The line numbers mentioned below refer to those in the revised manuscript. Point by point responses to editor/reviewers:

Reviewer #3:

We are very grateful to the Reviewer #1 for giving us very professional and specific suggestions to revise our manuscript.

Q1. It seems data were not analyzed statistically.

Thanks for your suggestion. We have supplemented the Statistical analysis.

At line 347-350: The activity of M-CPA and S-CPA were expressed as mean ± standard deviation of three independent experiments. The SPSS 20.0 statistical package was used to perform one-way analysis of variance (ANOVA), and Duncan's multiple range test was used to determine the significantly different (p < 0.05) between multiple samples. The Origin 7.5 software was used for drawing.

Q2. The data of M-CPA production rate (efficiency) by P. pastoris should be added into the section 2.1 right after Fig. 1 as this information is important to the up-scale enzyme production.

Thanks for your suggestion. We have revised it in the manuscript.

At line 97-98: The protein secretion by the recombinant P. pastoris GS115/pPIC9K/M-CPA reached to 303.08 mg/L.

Q3. Line 112: How was the optimum temperature obtained? Please write detail method in the method section.

Thanks for your suggestion. We have revised it in the manuscript.

At line 295-299: The optimal temperature of M-CPA and S-CPA were determined using the optimal pH at temperatures ranging from 20℃ to 80℃. The reaction solution was heated in water bath at different temperature for 10min, and then absorption value of this solution was tested by adding M-CPA or S-CPA. The calculation formula of relative enzyme activity was as follows: Relative activity = determined activity / activity at optimum temperature × 100%

Q4. Line 114-116: “The M-CPA and S-CPA retained 60.25% and 71.4% residual activity at temperature 40℃ for 10 min, respectively.” Why did this indicate that the thermal stability of M-CPA was higher than that of S-CPA?

Thanks for your suggestion. We have revised it in the manuscript.

At line109-120: Optimum temperature and thermal tolerance were used to characterize thermal stability of M-CPA and S-CPA. The optimum temperature of S-CPA (purchased from Sigma) and M-CPA (truncated expression) were determined during assay with 25 mM Tris-HCl buffer and 500 mM sodium chloride hippuryl-L-phenylalanine (HLP) (pH 7.5) as the substrate (Fig. 2 a). The optimum temperature of the S-CPA and M-CPA was found to be 30 ℃ and 40℃ respectively (Fig. 2 a). This indicated that the thermal stability of M-CPA was higher than that of S-CPA. The thermal tolerance of the M-CPA and S-CPA was carried out by incubating at different temperatures (30-80℃) for 10 min (Fig. 2 b). M-CPA and S-CPA retained 60.25% and 71.4% residual activity at temperature 40℃ for 10 min respectively. Above 50°C, the thermal tolerance of M-CPA was slightly better than that of S-CPA. However, their activity decreased more than 60% at temperatures above 50 ℃ and the residual activity only maintained 10% when the temperature above 70 ℃. These suggested that the thermal stability of M-CPA and S-CPA were both poor.

Q5. Line 121-122: “the M-CPA showed optimum pH within the range of 5–8.” should be changed to the M-CPA showed high enzymatic activity within the pH range of 5–8.

Thanks for your suggestion. We have revised it in the manuscript

Q6. Line 160: is it table1 ? The data in the table should be statistically analyzed.

Thanks for your suggestion. We totally agree with this comment. Two replicates were made at the in vitro simulated digestion assay, one for TLC detection and the other for HPLC detection. Table1 did not do statistical analysis, but this data can be used as a reference for colleagues. Therefore, we decided to adjust this part of the data to supplementary materials. We will do further application research on M-CPA in the follow-up. Then we will improve this assay at that time.

Q7. Line 209: Please change “stored” to “maintained “

Thanks for your suggestion. We have revised it in the manuscript.

Q8. Line274: Please specify the temperature for “the non-heated enzyme”.

Thanks for your suggestion. We have added the temperature in the manuscript.

At line 301-302: the non-heated enzyme (activity of enzyme in 25℃ with substrate) was used as the control (100%).

Q9. Line 309: Just after line309, please add method for detoxification in red wine.

Thanks for your suggestion. We have revised it in the manuscript.

At line 339-345: 5.10. Detoxification of OTA in red wine

To evaluate if M-CPA was able to biodegrade OTA in red wine, OTA was supplemented into red wine. OTA standard was added into 300 μL red wine samples with the final concentration of 2 μg/mL. There are three groups, containing experimental group (5U of S-CPA), control group (5U of S-CPA) and blank group. After the tubes were shaken and evenly mixed, they were incubated in a 200 rpm/min shaker at 37 ℃ for 24 h. The detoxification effect was detected according to the method in section 5.8.1, and the detoxification rate of recombinant CPA in wine samples was determined according to the method in section 5.8.2.

Q10. Line 318: Just after line 318, please add statistic data analysis

Thanks for your suggestion. We have supplemented the Statistical analysis.

At line 347-350: The activity of M-CPA and S-CPA were expressed as mean ± standard deviation of three independent experiments. The SPSS 20.0 statistical package was used to perform one-way analysis of variance (ANOVA), and Duncan's multiple range test was used to determine the significantly different (p < 0.05) between multiple samples. The Origin 7.5 software was used for drawing.

Reviewer 4 Report

Lines 13-14: “the recombinant M-CPA had an improve stability” →“ the recombinant M-CPA had an improved stability”

Line 81: “The recombinant M-CPA containing six histidine residues in their N-terminal region” → It contradicts with “A 6×His tag was introduced to the C-terminus of the target protein” in Line 246.

Line 83: “histidine-tagged lipases bound to the Ni-NTA resin was eluted” →“histidine-tagged carboxypeptidase bound to the Ni-NTA resin was eluted”

Lines 111-112: “sodium chloride HLP (pH 7.5)” → Please provide the full name of HLP when it first appears in the text.

Lines 127-128: “Compared with the S-CPA enzyme, approximately a 19% decrease in the specific activity.” → “Compared with the S-CPA, M-CPA was approximately a 19% lower in the specific activity.”

Lines 129-130: “indicated that the truncation of propeptide at the N-terminal region of CPA led to a substantial decrease slightly” → This sentence needs to be rephrased.

Figure 3: Why Figure 3b does not include data of 20℃ ?

Line 132: “Figure 3. Enzymatic properties of recombinant M-CPA. (a)The optimum temperature of M-CPA; (b)The thermal stability of M-CPA; (c)The optimum pH of M-CPA; (d) Enzymatic dynamic response curve of M-CPA.” → “Figure 3. Enzymatic properties of recombinant M-CPA and S-CPA. (a)The optimum temperature of M-CPA and S-CPA; (b)The thermal stability of M-CPA and S-CPA; (c)The optimum pH of M-CPA and S-CPA; (d) Enzymatic dynamic response curve of M-CPA and S-CPA.”

Line 145: “Figure 4. TLC and HPLC analysis of OTA biodegradation in red wine by recombinant M-CPA.” → “Figure 4. TLC and HPLC analysis of OTA biodegradation in red wine by recombinant M-CPA and S-CPA.”

Line 146: “2.5. M-CPA effectively degraded OTA in vitro gastrointestinal process” → “2.5. M-CPA effectively degraded OTA in vitro”

Line 203: “the recombinant M-CPA had an improve stability” →“ the recombinant M-CPA had an improved stability”

Lines 254-256: Why is the buffer B composition (pH 8.0, 50 mM NaH2PO4, 300 mM NaCl and 10 mM imidazole) the same as that (pH 8.0, 50 mM NaH2PO4, 300 mM NaCl and 10 mM imidazole) of buffer C?

Lines 265-267: Please specify the reaction time for the enzyme activity measurement.

Lines 270-271: “varying pH values (50 mM citrate buffer, pH 4.0–6.0; 50 mM PBS, pH 6.0–8.0; 50 mM Tris-HCl 270 buffer, pH 8.0–9.0; and 50 mM glycine-NaOH buffer, pH 9.0–10.0)” → Buffer pH ranges do not cover pH below 4.0 or above 10.0

Lines 264-277: The doses of M-CPA and S-CPA were not described. Please also describe how enzyme activity was measured (conditions and incubation time) after exposed to different temperature and different pH in details.

Lines 279-280: Please explain why the degradation assay has to been run for 24 hours. Were data of Figure 5 and Table 1 results of 24 hour incubation?

Line 280: Please define the unit for CPA enzyme activity.

Lines 280-281: “Assays with 5U/mL of carboxypeptidase A (EC 3.4.17.1) and blanks without any enzyme were used as controls.” → This is totally different from Lines 313-314: “For the gastric stage, 1.8 mL of gastric juice was added to the tube sets containing OTA (2 μg/mL) and M-CPA (10 U/mL),”

Lines 278-288: The treatment and dose of S-CPA were not described.

Lines 287-288: “The method of TLC analysis was the same as that of Shi [13].” → Please describe key steps of TLC analysis.

Lines 295-297: Please describe all the details how the red wine experiment was done include the treatment and dose of S-CPA.

Lines 307-318: The treatment and dose of S-CPA were not described.

Author Response

[Toxins] Manuscript ID: toxins-942214

The line numbers mentioned below refer to those in the revised manuscript. Point by point responses to editor/reviewers:

Reviewer #4:

We are very grateful to the Reviewer #1 for giving us very professional and specific suggestions to revise our manuscript.

Q1. Lines 13-14: “the recombinant M-CPA had an improve stability” →“ the recombinant M-CPA had an improved stability”

Thanks for your suggestion, we have revised it in the manuscript.

Q2. Line 81: “The recombinant M-CPA containing six histidine residues in their N-terminal region” → It contradicts with “A 6×His tag was introduced to the C-terminus of the target protein” in Line 246. Line 83: “histidine-tagged lipases bound to the Ni-NTA resin was eluted” →“histidine-tagged carboxypeptidase bound to the Ni-NTA resin was eluted”

Thanks for your suggestion. We have revised it in the manuscript. This is a write error; the six histidine residues are in their N-terminal region.

At line 98-101: The recombinant M-CPA containing six histidine residues in their N-terminal region was then purified using metal affinity chromatography on a Ni-NTA resin column. Each of these histidine-tagged carboxypeptidase bound to the Ni-NTA resin was eluted with imidazole.

Q3. Line 83: “histidine-tagged lipases bound to the Ni-NTA resin was eluted” →“histidine-tagged carboxypeptidase bound to the Ni-NTA resin was eluted”

Thanks for your suggestion, we have revised it in the manuscript.

Q4. Lines 111-112: “sodium chloride HLP (pH 7.5)” → Please provide the full name of HLP when it first appears in the text.

Thanks for your suggestion, we have revised it in the manuscript. The full name is hippuryl-L-phenylalanine (HLP).

Q5. Lines 127-128: “Compared with the S-CPA enzyme, approximately a 19% decrease in the specific activity.” → “Compared with the S-CPA, M-CPA was approximately a 19% lower in the specific activity.”

Thanks for your suggestion. We have rewritten the analysis of kinetic parameters.

At line 129-134: The Michaelis constant Km values of M-CPA and S-CPA hydrolyzing HLP were 0.126 mmol/L and 0.102 mmol/L respectively; the maximum rate (Vmax) of HLP hydrolysis were 0.0219 mol/min and 0.0315 mol/min respectively. These results indicated that the substrate affinity of M-CPA for HLP was slightly lower than that of S-CPA. The deletion of propeptide of CPA at the N-terminal region has a marginal effect on specific activity.

Q6. Lines 129-130: “indicated that the truncation of propeptide at the N-terminal region of CPA led to a substantial decrease slightly” → This sentence needs to be rephrased.

Thanks for your suggestion, we have revised it in the manuscript. These results indicated that the truncation of propeptide at the N-terminal region of CPA led to a substantial decrease slightly.

Q7. Figure 3: Why Figure 3 b does not include data of 20℃ ?

The purpose of this experiment is to investigated their thermal stabilities. The temperature of 20℃ is relatively low and have little application meaning.

Q8. Line 132: “Figure 3. Enzymatic properties of recombinant M-CPA. (a)The optimum temperature of M-CPA; (b)The thermal stability of M-CPA; (c)The optimum pH of M-CPA; (d) Enzymatic dynamic response curve of M-CPA.” → “Figure 3. Enzymatic properties of recombinant M-CPA and S-CPA. (a)The optimum temperature of M-CPA and S-CPA; (b)The thermal stability of M-CPA and S-CPA; (c)The optimum pH of M-CPA and S-CPA; (d) Enzymatic dynamic response curve of M-CPA and S-CPA.”

Thanks for your suggestion, we have revised it in the manuscript.

Q9. Line 145: “Figure 4. TLC and HPLC analysis of OTA biodegradation in red wine by recombinant M-CPA.” → “Figure 4. TLC and HPLC analysis of OTA biodegradation in red wine by recombinant M-CPA and S-CPA.”

Thanks for your suggestion, we have revised it in the manuscript.

Q10. Line 146: “2.5. M-CPA effectively degraded OTA in vitro gastrointestinal process” → “2.5. M-CPA effectively degraded OTA in vitro”

Thanks for your suggestion, we have revised it and this section has been adjusted to supplementary materials.

Q11. Line 203: “the recombinant M-CPA had an improve stability” →“ the recombinant M-CPA had an improved stability”

Thanks for your suggestion, we have revised it in the manuscript.

Q12. Lines 254-256: Why is the buffer B composition (pH 8.0, 50 mM NaH2PO4, 300 mM NaCl and 10 mM imidazole) the same as that (pH 8.0, 50 mM NaH2PO4, 300 mM NaCl and 10 mM imidazole) of buffer C?

There is a written mistake. We have revised it. Buffer C contains 50 mM NaH2PO4, 300 mM NaCl and 250 mM imidazole.

Q13. Lines 265-267: Please specify the reaction time for the enzyme activity measurement.

we have added the t reaction time in the manuscript.

Q14. Lines 270-271: “varying pH values (50 mM citrate buffer, pH 4.0–6.0; 50 mM PBS, pH 6.0–8.0; 50 mM Tris-HCl 270 buffer, pH 8.0–9.0; and 50 mM glycine-NaOH buffer, pH 9.0–10.0)” → Buffer pH ranges do not cover pH below 4.0 or above 10.0

Thanks for your suggestion, we have revised it in the manuscript.

At line 292-294: The optimal reaction pH was assessed using several buffers with varying pH values (50 mM acetic acid-sodium acetate buffer, pH 3.0-5.0; 50 mM NaHPO4-NaH2PO4 buffer, pH 6.0-7.5; 50 mM Tris-HCl buffer, pH 8.0-9.0; and 50 mM Gly-NaOH buffer, pH 9.0-12.0) at 25 ℃.

Q15. Lines 264-277: The doses of M-CPA and S-CPA were not described. Please also describe how enzyme activity was measured (conditions and incubation time) after exposed to different temperature and different pH in details.

Thanks for your suggestion, we have revised it in the method section manuscript.

At line 283-307:

5.7. Enzymatic Activity Measurements

The enzyme activity determination refers to the method of Tardioli et al [33]. The initial reaction rate was measured using hippuryl­L­phenylalanine (Sigma) as a substrate. The assay mixture contained the recombinant M-CPA dissolved in the 25 mM Tris-HCl buffer and 500 mM sodium chloride Hippuryl-L-phenylalanine (pH 7.5). Mix by inversion and record the increase in absorbance at 254 nm for approximately 5 minutes in a spectrophotometer. Obtain the fastest linear rate (ΔA254 nm/minute) over a 30 s interval for the test and the blank reactions. The initial enzymatic reaction rate was estimated from the linear region of the absorbance versus time curve. One unit of the enzymes will hydrolyze 1.0 μmole of hippuryl­L­phenylalanine per min at pH 7.5 at 25℃.

The optimal reaction pH was assessed using several buffers with varying pH values (50 mM acetic acid-sodium acetate buffer, pH 3.0-5.0; 50 mM NaHPO4-NaH2PO4 buffer, pH 6.0-7.5; 50 mM Tris-HCl buffer, pH 8.0-9.0; and 50 mM Gly-NaOH buffer, pH 9.0-12.0) at 25 ℃.

The optimal temperature of M-CPA and S-CPA were determined using the optimal pH at temperatures ranging from 20℃ to 80℃. The reaction solution was heated in water bath at different temperature for 10min, and then absorption value of this solution was tested by adding M-CPA or S-CPA. The calculation formula of relative enzyme activity was as follows: Relative activity = determined activity / activity at optimum temperature × 100%

The thermal tolerance assays were assessed by incubating the enzyme (M-CPA or S-CPA) at different temperatures (20℃ to 80℃) for 10 min, and the non-heated enzyme (activity of enzyme in 25℃ with substrate) was used as the control (100%). The calculation formula of relative enzyme activity was as follows: Relative activity = determined activity / activity of non-heated enzyme × 100%

The Km and Vmax values were estimated adjusting experimental kinetics data for substrate concentration from 0 to 1 mmol L-1 to Michaelis–Menten model with Origin 7.5 software package (OriginLab, Northampton, MA, USA)

Q18. Lines 279-280: Please explain why the degradation assay has to been run for 24 hours. Were data of Figure 5 and Table 1 results of 24 hour incubation?

There are many substrates of CPA, and OTA is one of its substrates. However, CPA has a low substrate affinity for OTA. It takes at least 24 hours for 5U CPA to completely degrade 2 ug of OTA, so we choose 24 hours to determine the degradation rate of OTA.

Q19. Line 280: Please define the unit for CPA enzyme activity.

Thanks for your suggestion. We have revised it in the manuscript.

Definition of CPA enzyme activity unit: One unit will hydrolyze 1.0 μmole of hippuryl­L­phenylalanine per min at pH 7.5 at 25ºC.

Q20. Lines 280-281: “Assays with 5U/mL of carboxypeptidase A (EC 3.4.17.1) and blanks without any enzyme were used as controls.” → This is totally different from Lines 313-314: “For the gastric stage, 1.8 mL of gastric juice was added to the tube sets containing OTA (2 μg/mL) and M-CPA (10 U/mL),”

Thanks for your suggestion. The two systems are of different sizes, so different amounts of enzyme are used.

Q21. Lines 278-288: The treatment and dose of S-CPA were not described.

Thanks for your suggestion. We have revised it in the manuscript.

At 309-316: 5.8. Ochratoxin A Degradation Assay

Degradation assays were done by incubating of recombinant M-CPA (5 U/ml) which dissolved in the 1 M NaCl (pH 7.5) with ochratoxin A solution at 37°C for 24 h. The final concentration of ochratoxin A was 2 μg/mL. Assays with 5 U/mL of S-CPA and blanks without any enzyme were used as controls. The aqueous phase was acidified to pH=2 and OTA was re-extracted three times with an equal volume of chloroform after centrifugation at 6000×g for 10 min, the organic phase was transferred to a clean tube, evaporated under nitrogen, dissolved in 300 μL methanol. The remaining OTA in the supernatants were applied to thin layer chromatography (TLC) analysis and analyzed by High-performance liquid chromatography (HPLC).

Q22. Lines 287-288: “The method of TLC analysis was the same as that of Shi [13].” → Please describe key steps of TLC analysis.

Thanks for your suggestion. We have revised it in the manuscript.

At line 318-321: 5.8.1 Thin layer chromatography analysis

The method of TLC analysis was performed according to Shi et al [23]. 10 μL supernatants of OTA were spotted on silica gel plates. Toluene/ethyl acetate/formic acid (6:3:1 v/v/v) was used as developing solvent. After 10 min of separating, OTA was chromogenic by the UV light of gel imaging analyzer.

Q23. Lines 295-297: Please describe all the details how the red wine experiment was done include the treatment and dose of S-CPA.

Thanks for your suggestion, we have revised it in the method section manuscript.

At line 339-345: 5.10. Detoxification of OTA in red wine

To evaluate if M-CPA was able to biodegrade OTA in red wine, OTA was supplemented into red wine. OTA standard was added into 300 μL red wine samples with the final concentration of 2 μg/mL. There are three groups, containing experimental group (5U of M-CPA), control group (5U of S-CPA) and blank group. After the tubes were shaken and evenly mixed, they were incubated in a 200 rpm/min shaker at 37 ℃ for 24 h. The detoxification effect was detected according to the method in section 5.8.1, and the detoxification rate of recombinant CPA in wine samples was determined according to the method in section 5.8.2.

Reviewer 5 Report

The aim of this study was to obtain recombinant mature carboxypeptidase A (M-CPA), the ability to efficiently and stably degrade OTA

The choice of topic is interesting and takes up a new issue.

The work was well planned and described in a manuscript that is very easy to read and follow. In my opinion, it is recommended for publication after some minor corrections:

A few punctuation problems are present in the manuscript. I suggest the Authors to double-check the text.

Author Response

Manuscript ID: toxins-942214

Reviewer #5:

Q1: The aim of this study was to obtain recombinant mature carboxypeptidase A (M-CPA), the ability to efficiently and stably degrade OTA

The choice of topic is interesting and takes up a new issue.

The work was well planned and described in a manuscript that is very easy to read and follow. In my opinion, it is recommended for publication after some minor corrections:

A few punctuation problems are present in the manuscript. I suggest the Authors to double-check the text.

Thank the reviewer 5# for affirming our work. We revised the manuscript carefully.

Round 2

Reviewer 2 Report

In the current revised submission of the manuscript entitled “Truncated expression of a carboxypeptidase A from bovine improves its enzymatic properties and detoxification efficiency of Ochratoxin A”, the authors have satisfactorily addressed most of the issues and concerns. In general, I am satisfied with the new version and authors’ replies to my specific comments. Unfortunately, the authors have not provided detailed information about the red wine used in the study. In my opinion, this was necessary if considering that possible interactions between wine compounds and OTA could produce OTA reduction besides the M-CPA action.

Author Response

Review 2#:

Comments and Suggestions for Authors

In the current revised submission of the manuscript entitled “Truncated expression of a carboxypeptidase A from bovine improves its enzymatic properties and detoxification efficiency of Ochratoxin A”, the authors have satisfactorily addressed most of the issues and concerns. In general, I am satisfied with the new version and authors’ replies to my specific comments. Unfortunately, the authors have not provided detailed information about the red wine used in the study. In my opinion, this was necessary if considering that possible interactions between wine compounds and OTA could produce OTA reduction besides the M-CPA action.

Thanks for the suggestion. We complement some raw material information of the red wine. The grape variety used to make this wine was Cabernet Sauvignon, producing in Helanshan, Ningxia, China (38°34'N, 106°20'E). The wine was brewed in 2017, with an alcohol content of 13.4% and a pH of 3.52.

We believe that the main factor affecting the degradation rate of CPA to OTA in red wine is pH. As shown in Fig. 2c, S-CPA has no OTA degradation activity below pH 4.0, while the OTA degradation rate of M-CPA still retains 65%. And the red wine pH of this study was 3.54.

We agree that during the wine production process, certain components of red wine, such as anthocyanins, may interact with OTA to form some OTA derivatives and reduce the content of OTA. Studies have reported that phenolics can reduce the toxicity of OTA to cells and animals. The phenolics of red wine may interact with OTA to form new OTA derivatives. The toxicity of these derivatives requires further safety evaluation research. Therefore, the toxicity evaluation of OTA in red wine should not only consider the reduction of its content but also comprehensively evaluate the toxicity of other derivatives. We will consider this issue in subsequent safety evaluation research.

Reviewer 4 Report

Lines 114-116: “The thermal tolerance 114 of the M-CPA and S-CPA was carried out by incubating at different temperatures (30-80℃) for 10 min (Fig. 2 b).” which is not what stated in Lines 300-301 “The thermal tolerance assays were assessed by incubating the enzyme (M-CPA or S-CPA) at different temperatures (20 to 80℃) for 10 min”.

Figure 2. (b) should present the data of 20℃ because in Lines 300-301 “The thermal tolerance assays were assessed by incubating the enzyme (M-CPA or S-CPA) at different temperatures (20 to 80℃) for 10 min” which has clearly indicated thermal tolerance assays was done for 20℃.

Lines 128-131: Please explain how Km values of M-CPA and S-CPA (0.126 mmol/L and 0.102 mmol/L) and the maximum rate (Vmax) (0.0219 mol/min and 0.0315 mol/min) can be obtained from Figure 2. (d). 

Lines 305-307: Please specify how many substrate concentrations were tested.

Author Response

Review 4#:

Comments and Suggestions for Authors

Q1:Lines 114-116: “The thermal tolerance 114 of the M-CPA and S-CPA was carried out by incubating at different temperatures (30-80℃) for 10 min (Fig. 2 b).” which is not what stated in Lines 300-301 “The thermal tolerance assays were assessed by incubating the enzyme (M-CPA or S-CPA) at different temperatures (20 to 80℃) for 10 min”. Figure 2. (b) should present the data of 20℃ because in Lines 300-301 “The thermal tolerance assays were assessed by incubating the enzyme (M-CPA or S-CPA) at different temperatures (20 to 80℃) for 10 min” which has clearly indicated thermal tolerance assays was done for 20℃.

Thanks for the suggestion. The thermal tolerance assays were carried out at the temperatures of 30℃ to 80℃. It has been corrected in the text.

In lines 300-301: The thermal tolerance assays were assessed by incubating the enzyme (M-CPA or S-CPA) at different temperatures (30℃ to 80℃) for 10 min.

Q2:Lines 128-131: Please explain how Km values of M-CPA and S-CPA (0.126 mmol/L and 0.102 mmol/L) and the maximum rate (Vmax) (0.0219 mol/min and 0.0315 mol/min) can be obtained from Figure 2. (d). Lines 305-307: Please specify how many substrate concentrations were tested.

Thanks for the suggestion. In line 305-310, we supplement the calculation of Km and Vmax.

Km and Vmax values were estimated by the Michaelis–Menten model, basing on the experimental kinetics data for substrate concentration at 0.2, 0.4, 0.6, 0.8, and 1 mmol/L. Km and Vmax can be obtained according to the Lineweaver-Burk plot. It gives a straight line, with the intercept on the y-axis equal to 1/Vmax, and the intercept on the x-axis equal to the absolute value of 1/Km. The calculation formulas of Km and Vmax are as follows, where [S] refers to the concentration of the substrate.
